# Highlighting the Microbial Contamination of the Dropper Tip and Cap of In-Use Eye Drops, the Associated Contributory Factors, and the Risk of Infection: A Past-30-Years Literature Review

**DOI:** 10.3390/pharmaceutics14102176

**Published:** 2022-10-12

**Authors:** Katia Iskandar, Loïc Marchin, Laurent Kodjikian, Maxime Rocher, Christine Roques

**Affiliations:** 1Département des Sciences Pharmaceutiques, Faculté de Pharmacie, Université Libanaise, Beirut 1500, Lebanon; 2INSPECT-LB—Institut National de Santé Publique, d’Épidémiologie Clinique et de Toxicologie-Liban, Beirut 1202, Lebanon; 3Pylote SAS, 22 Avenue de la Mouyssaguèse, Dremil-Lafage, 31280 Toulouse, France; 4Service d’Ophtalmologie, CHU de la Croix-Rousse, Hospices Civils de Lyon, 69004 Lyon, France; 5Département d’Ophtalmologie, Université de Lyon 1, UMR-CNRS 5510, Matéis, INSA, Villeurbanne, 69100 Lyon, France; 6Department of Ophthalmology, Limoges University Hospital, 87000 Limoges, France; 7Institut National de la Santé et de la Recherche Médicale (INSERM), Centre Hospitalier Universitaire Limoges, Université de Limoges, RESINFIT, 87000 Limoges, France; 8Laboratoire de Génie Chimique, Université de Toulouse, CNRS, INPT, UPS, Faculté de Pharmacie, 31062 Toulouse, France; 9FONDEREPHAR, Faculté de Pharmacie, 31062 Toulouse, France

**Keywords:** eye drops, dropper tip, cap, microbial contamination, ocular infection

## Abstract

The sterility of eye drop content is a primary concern from manufacturing until opening, as well as during handling by end users, while microbial contamination of the dropper tip and cap are often disregarded. The contamination of these sites during drug administration represents a risk of microbial transmission and ocular infection. In this review, we aim to assess microbial contamination of the dropper tip and cap of in-use eye drops, the associated contributory factors, and the risk of infection. We conducted a literature search of the MEDLINE, PubMed, and Cochrane Central databases. A total of 31 out of 1503 studies were selected. All the studies conducted in different settings that documented microbiologically contaminated in-use eye drops were included. Our review showed that microbial contamination of the dropper tip and cap of in-use eye drops ranged from 7.7 to 100% of the total contaminated tested samples. Documented contributory factors were conflicting across the literature. Studies investigating the association between eye infection and microbial contamination of the dropper tip and cap were scarce. New technologies offer a promising potential for securing the long-term sterility of eye drop content, tips, and caps, which could benefit from more research and well-defined study protocols under real-life scenarios.

## 1. Introduction

The eye is a complex organ with specialized anatomy and physiology [1,2,3,4] and a robust local immune response [5,6]. A wide range of pathologies can affect the eyes due to local triggers or systemic diseases [7]. Therapeutic and diagnostic eye drops offer a noninvasive route of ocular drug delivery [1]. Eye drops are sterile topical ophthalmic formulations containing a solution, emulsion, or suspension of one or more active ingredients [8]. These conventional dosage forms account for 90% of marketed ophthalmic solutions [1,2]. Eye drops are generally ready-to-use marketed products with or without preservative agents [8] or, to a lesser extent, prepared in hospitals to meet specific patients’ needs [8,9,10]. Eye drops are available in unit-dose and multiple-dose containers (Figure 1).

The sterility of eye drops should be, ideally, secured throughout the manufacturing process and the supply chain and maintained for the recommended duration of use, from opening to handling by the end user [10,11]. A literature review showed that the microbial contamination rate of preserved and preservative-free eye drops varied from 2.3 to 73% [12,13,14,15,16,17,18,19,20,21,22,23,24,25,26,27,28,29,30,31,32,33,34,35,36,37,38,39,40,41,42,43,44,45,46,47,48,49,50]. The most important point was that fungal and bacterial microorganisms were cultured from the dropper tip, cap, and contents of eye drops [20,22,26,39,43,46,49]. Bacteria from the human commensal flora were the predominant isolated microorganisms [17,18,21,22,26,30,34,41,43,46,49], logically, since the contamination occurred after the bottle was opened. Limited growth of pathogenic bacteria and airborne environmental bacterial and fungal spores were also reported [15,24,29,34,43,50].

Microbial contamination of in-use eye drops is a risk of ocular infection and a leading cause of potentially avoidable eye injury [45,50,51,52,53,54,55]. Studies have shown that microbial contamination of the dropper tip and cap is associated with ocular injuries, such as corneal injury [52,56] and bacterial keratitis [45]. The risk of ocular infection is considered high in cases of extensive contact lens wear, ocular trauma, recent eye surgery, preexisting ocular diseases, lid deformity, and extended use of topical steroids, systemic steroids, and immunosuppressants [18,19,21,22,23,30,34,35,37,43,49]. A literature review showed a limited number of studies that examined in-depth the link between ocular infection and the microbial contamination of eye drops. Microbial contamination was documented when used by patients at home and by healthcare professionals (HCP) in private clinics, in inpatient and outpatient settings, and in the operating room (OR) [15,17,18,19,36,37,39,41,43,49]. Numerous studies have reported inconsistent findings attributable to contributory factors [15,17,18,19,21,23,24,37,46,47,49,50]. The handleability of eye drops may pose an important risk of microbial contamination [10]. The squeezability factors [11], missing the eyes completely during drug instillation [10], variable duration of use (preserved versus preservative-free [2,3,6,8,12,34,46,50], improper administration techniques, and age-related physical difficulties [17,24,32,37,47,49] were potential causes of microbial contamination at the dropper tip. Schein (1992) [41] proposed a cycle of eye drops and eye contamination. During the opening, recapping, and instillation, the dropper tip may inadvertently touch the ocular surface and cilia [24,37,41,43,49,50]. The contact of the dropper tip and eye or ocular annexes during drug administration may not only lead to the contamination of the former but also may contribute to cross-contamination with pathogenic microorganisms originating from an infected patient and affecting another patient with disrupted epithelial barriers or a compromised corneal epithelium using the same eye drops [43]. Pharmaceutical companies are striving to find alternatives to secure the sterility of the eye drop content, while the tip and cap have not seemed to be a priority until recent years. In this review, we aim to assess microbial contamination of the dropper tip and cap of in-use eye drops, the associated contributory factors, and the risk of infection.

## 2. Materials and Methods

A literature search for the identification of relevant studies was conducted on May 1, 2022, using the below three electronic databases:Ovid MEDLINE(R) Epub Ahead of Print, In-Process, and Other Non-Indexed Citations; Ovid MEDLINE(R) Daily; and Ovid MEDLINE(R) 1946 to Present;PubMed (http://pubmed.gov, accessed on 10 May 2022);Cochrane CENTRAL.
The context of this review was the evaluation of in-use eye drop tip and cap documented microbial contamination regarding hospital settings (inpatient and outpatient clinics, operating rooms), long-term facilities, primary care clinics, and home-based settings.The search strategy principle was based on dividing the topic into two concepts: (1) eye drops and (2) microbial contamination. Ovid MEDLINE was first searched to identify all the possible MeSH terms with their corresponding keyword equivalences to increase the sensitivity of the search strategy. The search strategy combined the two concepts as follows: exp Ophthalmic Solutions/OR ((ophthalm* or ocular or eye?) adj2 (MEDICATION? or drop? or solution? or preparation?)).tw. OR (((eye or ophthalm* or ocular) adj2 drop?) or eyedrop?).tw. AND exp Eye Infections/AND Drug Contamination/AND (contamination? or cross-contamination?).tw. [mp = title, abstract, original title, name of substance word, subject heading word, floating subheading word, keyword heading word, organism supplementary concept word, protocol supplementary concept word, rare disease supplementary concept word, unique identifier, synonyms]. All searches were limited to humans and the English language, with restrictions on publication dates starting from 1 January 1992 to 1 June 2022.After finalizing the MEDLINE strategy, the search terms were appropriately adapted to the other two databases. The search results were exported into Zotero (https://www.zotero.org/) (accessed on 1 June 2022) to remove duplicates. The citations were then imported to Rayyan (https://www.rayyan.ai/) (accessed on 10 June 2022) to screen the articles.The studies were excluded if their primary objective was not solely in-use eye drops documented with microbial contamination.Further studies were identified by a hand search and by examining the reference lists of all the included articles.Data extraction from the selected publications focused on descriptive, quantitative, and microbiological results. We only included full-text articles. The extracted information was the location and year of publication, study design, objective, setting, inpatient or outpatient use, description of ophthalmic solutions tested, single- or multi-dose eye drop, with or without preservatives, type of eye drops, type of preservative, intended use, duration of use, rate of contamination, eye drop site of contamination, outcomes description, and relevant findings, including the source of contamination and isolated microorganisms.A total of 31 publications were included in this review (Figure 2).Tables and figures are available for the databases search strategy

## 3. Results

Out of 31 studies included in the review, 15 were conducted in high-income countries according to the World Bank classification of countries by income level. (https://datahelpdesk.worldbank.org/knowledgebase/articles/906519-world-bank-country-and-lending-groups) (Accessed 15 June 2022). A total of 64% of the studies took place in hospitals in one or more settings, including OR (20%), outpatient (75%), and inpatient settings (55%). Most of the study designs were descriptive, cross-sectional, and observational research (Table 1).

### 3.1. Microbial Contamination of Eye Drops According to the Setting of Use

#### 3.1.1. In-Use Eye Drops Collected from Inpatient and Outpatient Settings

The microbial contamination rates of preserved in-use eye drops collected from inpatient and outpatient settings varied widely. Four studies reported that the microbial contamination rates of multi-use therapeutic or diagnostic eye drops collected from outpatient settings were 11.7% [19], 15% [39], 25.45% [15], 30% [18] and 72.8% [43], with the highest percentage noted on day fourteen [15,18,43]. Other studies collected in-use diagnostic eye drops from ophthalmology clinics and found a 94.46% microbial contamination rate [28]. Livingstone (1998) [30] tested eye drops retrieved from inpatient settings and found no significant difference in the microbial contamination rates between day seven (6.1%) and day fourteen (9.1%) [30]. Nentwich (2007) [34] evaluated the microbial contamination of eye drops used by single- and multi-users in different settings [34]. In this study, the results showed a 6% microbial contamination rate of 101 tested eye drops after fourteen days of use, with 5% (4/77 collected bottles) documented in multi-user settings versus 8% (2/24 collected bottles) from single-user settings contaminated with different bacterial strains [34]. Teuchner (2015) [46] evaluated the microbial contamination of eye drops after one week of use in the OR and compared the results among outpatient and inpatient units and home use after four weeks. The overall rate in the study was 17% [46]. The contamination rate in the OR was significantly lower, potentially due to the limited period of use compared with the other settings. Tsegaw (2017) [50] found an overall contamination rate of 11% in tested eye drops used by patients and HCP [50]. In this study, the bacterial contamination was different in eye drops used for less than one week (3.2%) compared with those used for more than one week (24.3%) [50]. The rate of contamination varied from 2.3% after ≤72 h and 2% after five days [21] to 8% after seven days [22] and completed use [42]. Feghhi (2008) [48] found a higher contamination rate on day one compared with day seven, with an overall rate of 18% [48] (Table 2).

The contamination rates differed in the studies that tested eye drops used for a week or less. The lowest rates were found in the OR in all studies. The wide variability in the findings may be related to the setting, the duration of use, the user(s), the institution’s eye drop use protocol, and the number of samples taken, in addition to the sampling site for culture (residual content, drops, tip, cap, thread) and the sampling method, as well as a microbial culture method that may not cover all microorganisms, such as anaerobes and fungi.

Other studies evaluating the microbial contamination of preservative-free eye drops were limited. The findings in these studies were not comparable since the type of medication, the design of the container, and the settings were different.

Chantra (2022) [17] collected antimicrobial multi-dose eye drops used for two or more days by the patients and HCP. The rate of contamination with bacterial and fungal microorganisms was 24.06%. The authors tested multiple sites of the eye drop container and showed that the tip was more contaminated (49.2%) than the residual content of the eye drops (26.7%) [17]. Mason (2005) [57] evaluated the microbial contamination of an ophthalmic moxifloxacin solution and had similar results, with a contamination rate of 1.6% [57]. The culture was positive at the thread of the container. The potential self-preserved effect of the antibiotic was considered the principal cause of the limited microbial contamination.

Rahman (2006) [37] tested multi-use preservative-free eye drops extracted from inpatient (on day three) and outpatient setting (on day seven) [37]. Eye drop formulations, including antimicrobials, steroids, Hypromellose, and cyclosporine, were prepared on-site in a hospital in a glass container with a pipette attached to the cap. The results showed that the bacterial contamination rate was 8.4% in non-antibiotic eye drops [37].

Thanathanee (2013) [14] tested preservative-free multi-dose 100% autologous serum eye drops prepared on-site in a hospital and administered to inpatients by trained medical personnel. The results showed 6.1% microbial contamination [14]. Similar studies of 20% autologous serum found 9.7% microbial contamination in hospital-based therapy [13] and 2.1% in containers with adapted filters compared with 28.9% microbial contamination of conventional containers in home-based therapy [12].

Somner (2010) [25] examined the contamination rate of the minims dropper tips used once in eye clinics according to the “no touch” technique and found a 5% bacterial contamination rate. The tested eye drops included phenylephrine, tropicamide, fluoresceine, and proxymetacaine [25] (Table 2).

#### 3.1.2. In-Use Eye Drops Collected from Patients

The microbial contamination rates of preserved eye drops used by patients were significantly higher than those used in inpatient and outpatient settings [43,46]. In these studies, the dropper tips were more contaminated than the drops and residual content [43,46]. Schein (1992) [41] found an overall 29% microbial contamination in eye drop bottles used by patients for at least two months. Geyer (1995) [49] tested glaucoma medications used by patients for at least three months. The rate of bacterial contamination was 28% [49]. Two similar studies found a microbial contamination rate of 9.1% when glaucoma eye drops were used for more than thirty days [24] and 12.9% after a period ranging from one to twelve weeks [36]. Porges (2004) [36] showed that the magnitude of microbial contamination differed between eye drops used for less than four weeks (19%) and those used for more than 12 weeks (40%) [36]. Another study tested therapeutic eye drops collected from patients after a maximum period of fourteen days of use and found a microbial contamination rate of 64.83% [29].

Kim (2008) [47] compared preservative-free (case) versus preserved artificial tears (control) in reclosable containers collected from patients after one day of use. The microbial contamination rate was 2% in the preservative-free medications [47] (Table 2).

The majority of these studies tested glaucoma medications and found different contamination rates, usually higher than those used in hospital settings, from about 10% to more than 60%. The variability in the findings may be related to differences in the number of tested samples, study designs, sample collection methods, and microbiological assays.

### 3.2. Contaminated Eye Drop Sites

Numerous studies took cultures from drops [22,37,42,47] and the residual content of eye drops [29,30]. Some other studies evaluated microbial contamination from multiple sites, such as the cap, dropper tip, drops, and residual content. The main findings indicated higher microbial contamination of the dropper tip compared with the residual content [17,18,20,21,26,34,35,36,39,43,45,46,48,49,50] (Table 2). Tamrat (2019) [43] showed that the tips of tested eye drops taken from inpatient, operating room, and outpatient settings were 61% more contaminated than the residual content at 4%. Similar studies conducted in outpatient settings obtained the same results [18,39], even if differences in the microbial contamination rates between the tip and the residual content varied across different studies. Geyer (1995) [49] collected glaucoma medications containing BAK from patients and noted differences in the contamination between the tip (70.9%) and the content (29%). Teuchner (2015) [46] compared the microbial contamination of glaucoma eye drops used by patients versus the hospital staff. The results showed that microbial contamination of the tip was higher (20.2%) than those of the drops (8.4%) and the residual content (5%) when applied by patients. At the ward level, the dropper tip was the least contaminated [46]. The frequency of microbial contamination of the tip was lower (1.5%) than those of the drops (11.3%) and the residual content (7.5%) when applied by medical personnel [46].

Tsegaw (2017) [50] tested eye drops used by patients and HCP and obtained microbial growth at the tip but not in the residual content. Most ophthalmic solutions are characterized by low propensity for microbial growth, even if preservative-free, which may explain the low bioburden and “natural” decrease in the bioload of the residual content. Chua (2021) [18] studied the microbial contamination of eye drops in an ophthalmology clinic and found that the dropper tip was significantly more contaminated (50%) than the residual content (33%) [18]. Razooki (2011) [39] and Nenwitch (2007) [34] obtained the same results (tip: 62.5% versus residual content: 25% [39] and tip: 83.33% versus residual content: 16.6% [34], respectively). Fazeli (2004) [23] showed different findings where the residual drops were more contaminated than the tip [23].

Although multi-use eye drops contain a preservative, the contact time between the antimicrobial agent to exert its effect and the tip is limited and insufficient. The proposed mechanism of contamination of the content of eye drops from the dropper tip is microbial desiccation and aspiration of the content at the tip level [38].

Chantra (2022) [17] tested preservative-free multi-dose eye drops containing antimicrobials collected from inpatient and outpatient units. These medications were formulated on-site in the hospital setting. The results showed that the tip was 49.2% contaminated, the residual content 26.7%, while both sites were 17%. Surprisingly, the dropper tip of the eye drops was most contaminated when used by HCP [17]. Donzis (1997) [56] reported microbial contamination of the dropper tip only for a multi-dose preservative-free saline aerosol used by patients. Daehn (2021) [21] collected in-use eye drops from the operating theater of an ophthalmology hospital and found 2% contamination of the tip of the total tested sample, equivalent to 100% of the contaminated eye drops (five bottles), despite hygiene instructions in this setting.

Other studies showed higher contamination of the caps of ophthalmic solutions in private eye clinics [19] and hospital settings [57]. In these studies, the dry residues in the eye drop cap and thread may be the origin of the microbial contamination [19,57]. Feghhi (2008) [48] showed that eye drops used by patients were more contaminated at the cap (46%), followed by the tip (41%) and the residual content (13%), confirming that the caps of squeeze bottles serve as a reservoir for microbial contamination and then growth [48]. Other studies showed microbial contamination of the content or residue in eye drops taken from different settings [30,37,42,47]. The proposed mechanism of microbial contamination of the moist tip originating from the dead space in the cap is the transmigration of the microbial contaminants from the dropper tip, leading to contamination of the content [49] (Table 2).

Most studies did not report counts of colony-forming units (CFUs) from different contaminated sites of the eye drops. Teuchner (2015) [46] showed that the ratio of microbial contamination of human to environmental flora was 24/7 at the dropper tip compared with 5/10 found in the residual content and 12/14 in the drops. In this study, glaucoma eye drops collected from patients showed microbial contamination with *Staphylococcus* spp. of more than 1000 CFUs at the dropper tip, while in other collected samples, CFU counts higher than 50 were found in eye drops used at home (56 and 300 CFU) compared with much lower counts in the wards (2 and 6 CFU) and the OR (1 CFU) [46].

### 3.3. Types of Contaminated Eye Drops

#### 3.3.1. Types of Tested Eye Drop Medications

In-use therapeutic and diagnostic ocular formulations showed different levels of contamination. Mydriatic agents, anesthetics, glaucoma treatments, steroids, and antimicrobials eye drops were the most extensively studied ophthalmic medications. Other tested eye drops were lubricants, miotics, antihistamines, and nonsteroidal anti-inflammatory drugs (NSAIDs) (Table 3).

Chua (2021) [18] tested mydriatics and anesthetic medications manufactured by the same company and formulated with similar percentages of BAK. The results showed that proparacaine, the local anesthetic agent instilled by an ophthalmologist, was more contaminated than tropicamide, a mydriatic agent handled by a clinic assistant or nurse [18]. The higher level of microbial contamination detected in anesthetics was attributable to the frequency of use. These findings are consistent with similar studies [19,23,34,43,50]. In other studies, mydriatics were more contaminated than NSAIDs [15], lubricants [15], and glaucoma medications [28]. Steroid eye drops were additional highly tested medications. Jokl (2007) [22] showed that steroid eye drops were 5.8 times more contaminated than topical antimicrobial solutions alone or anti-inflammatory drugs, lubricants, mydriatics, miotics, and medications intended to treat glaucoma [22]. Numerous studies have reported the microbial contamination of steroids [23,34,39,41,43,48,51] compared with other therapeutic and diagnostic agents. Feghhi (2008) [48] found that pilocarpine, a miotic agent, was the most contaminated, followed by mydriatic agents and steroids. In this study, other tested eye drops, such as topical ocular antibiotics and beta-blockers, showed microbial contamination [48]. Geyer (1995) [49] found that medications used to treat glaucoma, mainly beta-blockers, were more contaminated than miotics, sympathomimetics, and other eye drops, such as steroids and antibiotics. Figuêiredo (2018) [24] obtained the same findings, where the beta-blocker timolol was contaminated the most among other glaucoma treatments. Teuchner (2015) [46] reported similar results, where glaucoma eye drops were more contaminated than anesthetics and antibiotics. Porges (2004) [36] showed that hypotensive eye drops had a low contamination rate when used by patients. The sampling method [36], sample size, and the tested sites may potentially explain the differences in findings if similar types of eye drops were compared. Overall, studies have demonstrated higher microbial contamination of therapeutic eye drops than diagnostic eye drops [22,24,41,49]. Tested eye drops containing antibiotics were either sterile [34,50] or less frequently contaminated than nonantibiotics [37,43]. This finding is attributable to the potential killing effect of antibiotics, keeping in mind various antibacterial spectra. In preservative-free eye drops, a study by Chantra (2022) [17] tested topical antimicrobial eye drops and found that vancomycin was the most frequently contaminated formulation. The study showed that 26 pathogens, predominantly molds, were isolated, in addition to GNB and GPB, such as *Staphylococcus aureus*, *Corynebacterium* spp., and *Pseudomonas aeruginosa* (42.98%) [17]. The study showed that 26 pathogens were isolated, including Gram-positive and Gram-negative bacteria and, predominantly, molds (42.98%) [17].

The literature did not consider whole formulations but rather the presence of preservative agents or not, which are known to impact the ability of microorganisms to survive and then proliferate. The tested ophthalmic solutions contained different types and concentrations of preservatives but also various active ingredients and excipients able to impair preservation. At the same time, the presented studies often had no controls and cultures taken from all possible sites of eye drops prone to microbial contamination, such as the dropper tip, cap, drops, and residual content.

#### 3.3.2. Preserved and Preservative-Free Eye Drops

##### Preserved Eye Drops

Preservatives are recommended additives to topical ophthalmic solutions that are intended to prevent the microbial contamination of eye drops [58,59,60,61,62]. The European Medicines Agency (EMA) supports the justified use of mercury-free antimicrobial preservatives in eye drops if formulated at the minimum effective concentration while optimizing the benefit–risk ratio [63,64]. There are three main types of preservatives in ophthalmic solutions: (1) detergents are expected to exert a broad spectrum of action by disrupting the lipid cell membrane layer, causing bacterial cell lysis [65,66]; (2) oxidative agents are second-generation ocular preservatives that act by penetrating bacterial cell membranes and damaging their DNA, proteins, and lipid components [66] (these preservatives are effective at low concentrations [67] and less harmful to the cornea than detergents [68]); and (3) ionic buffer systems have antibacterial and antifungal activities and similar mechanisms of action to oxidative agents [66]. These antimicrobial agents have shown a toxic effect on corneal and conjunctival surfaces with extended use [51,58,61,66,69].

Numerous studies have shown that eye drops containing preservatives can be, however, contaminated with bacteria, viruses, and fungi [15,18,19,21,23,24,30,34,35,36,39,48,49,52,70]. These observations are seen with detergents, such as BAK used at different concentrations (0.004 to 0.05%) [15,18,19,21,24,30,35,39,48,49], phenylmercuric nitrate (0.001–0.002%) [21,34], and chlorbutol IP (0.5%) [15,28,34], in addition to oxidative agents, such as thimerosal (0.001−0.005%) [34] and sodium perborate [28,29]. BAK was the most widely reported antimicrobial agent in most of the tested content of preserved ophthalmic solutions that showed microbial contamination [18,21,23,24,29,30,34,35,39,42,48,49,52,70]. Livingstone (1998) [30] showed that the contamination rates of ophthalmic solutions formulated with lower BAK concentrations (0.004–0.005%) were higher compared with those containing higher BAK concentrations (0.01–0.02%) [30].

Additional factors are the pH of a solution, storage conditions, and the physicochemical properties of the ingredients [36] (Table 3).

##### Benzalkonium Chloride Preservatives

BAK is the most extensively studied preservative [58,62], added to 70% of ophthalmic formulations [58,71]. BAK has documented clinical, experimental, and laboratory ocular toxicity inherent to its mechanism of action. The adverse effects attributable to the cytotoxic effects of BAK can lead to ocular surface diseases (OSDs), such as dry eye aggravated with chronic use [51,61]. The ocular toxicity of BAK may also manifest within seven days of exposure [51,58,72,73]. These effects can be totally or partially reversible upon BAK withdrawal [58,74,75,76,77]. BAK is more active against GPB than GNB and has no or poor efficiency on bacterial endospores, acid-fast bacteria, and fungi [78,79].

In 2017, the EMA set the labeling requirements for benzalkonium chloride based on its safety profile [80], predominantly in neonates [64]. Tested in-use eye drops containing BAK have been documented with microbial contamination in numerous studies [15,18,19,21,23,24,29,30,34,43,46,48,49,50,52,60]. Despite the controversial use of quaternary ammonium compounds as eye drop preservatives, BAK is still widely attractive to companies because it offers many advantages, which include enhancing the penetration of the active ingredient of ophthalmic solutions into the eyes, and is approved in all countries. BAK is cost-effective compared with new preservatives and practical to handle for patients [72,73] (Table 3).

##### Preservative-Free Eye Drops

In 2009, the European Medicines Agency (EMA) recommended preservative-free ophthalmic preparations for patients intolerant to preservatives in pediatrics, particularly neonates, as well as if long-term use of these drugs is needed [63]. The recommendations support the justified use of new formulations containing mercury-free antimicrobial preservatives added at the minimum effective concentration and the optimum benefit–risk ratio [62,63]. Preservative-free eye drops are mainly commercially formulated in unit-dose vials or locally prepared on-site in hospitals and stored in multi-dose glass containers [37]. The limits of use of these last preparations are determined based on practical considerations rather than evidence-based ones. Even though poorly documented, these hospital-prepared formulations presented high risks of contamination, ranging between 8.4% [37],16.7% [40], 24.06% [17], and 28.9% [12] (Table 3). Currently, new packaging designs are being evaluated and proposed to limit the risk of microbial migration to the eye-drop contents and the viability of contaminants on caps and tips [12,81,82,83].

### 3.4. Isolated Microorganisms

In numerous studies, Gram-positive bacteria (GPB) were the only type of bacteria isolated from the drops [42], threads [57], residual contents [23], caps [23], and dropper tips of in-use ophthalmic solutions [21]. The isolated microorganisms were predominantly *Coagulase-negative staphylococci* (CoNS) [21,23,42,57]. In three studies, most eye drops contained BAK [21,23,42].

Despite this, some other studies have reported sole microbial contamination with Gram-negative bacteria (GNB). Donzis (1997) [56] showed contamination of the dropper tip of preservative-free aerosol saline sprays with *Pseudomonas aeruginosa* [56]. Figuêiredo (2018) [24] evaluated the microbial contamination of hypotensive eye drops used by patients with glaucoma. The isolated bacteria were *P. aeruginosa*, *Serratia marcescens*, and *Stenotrophomonas maltophilia*, usually part of hospital opportunistic microflora. All patients using the contaminated eye drops stated that they had visited a hospital within the past thirty days of eye drop use [24], potentially explaining the source of these bacteria.

Similar studies that tested eye drops used by patients to treat glaucoma found microbial contaminants of GNB and GPB [22,36,49]. GPB were isolated from the dropper tip, while GNB were cultured from the dropper tip and the contents [36,49]. Geyer (1995) [49] found that GPB isolated from the conjunctiva were the same as those found in the tested eye drops. The results included CoNS, *diphtheroids*, *Propionibacterium (Cutibacterium) acne*, and *Streptococcus viridans*, resulting potentially from direct contact during instillation between the tip of the ocular medication and the eyelids [49]. The GNB were *Moraxella* spp., *Enterobacter aerogenes*, *Serratia marcescens*, *Flavobacterium* spp., *Pseudomonas* spp., and *Proteus mirabilis*; *P. mirabilis* was cultured from the conjunctiva of glaucoma patients [49]. Porges (2004) [36] found *Pseudomonas* spp., *Klebsiella* spp., and *Staphylococcus epidermidis* in the content and dropper tip of the bottle, while *S. viridans* was cultured from the content only. All the contaminated eye drops contained a high level of BAK (0.02%). In these two last studies [36,49], the tested eye drops were in use by patients and intended to treat glaucoma.

Jokl (2007) [22] studied bacterial contamination of the contents of different types of ophthalmic formulations in an extended care facility. The results showed predominant contamination by *P. mirabilis* and secondary by *Klebsiella pneumoniae*. The isolated GPB were *S. epidermidis*, *Clostridium perfringens*, and other Gram-positive cocci. Tsegaw (2017) [50] found that multi-use eye drops and extended duration of use were associated with the contamination of eye drops with CoNS and *Bacillus* spp., in addition to GNB such as *Escherichia coli* and *Enterobacter* spp. The tip of the eye drop bottle was the only contaminated site where the residual content of the eye drop bottles showed no microbial contamination. The isolated bacteria were *Staphylococcus aureus* methicillin-resistant [50]. Nisar (2018) [35] tested the residual contents of eye drops used by patients. The microbial contaminants were Gram-positive cocci and bacilli, while the GNB included *P. mirabilis* and *K. pneumonia* [35]. Chua (2021) [18] studied multi-dose eye drops used by HCP. The identified microorganisms were CoNS and *Micrococcus* spp., in addition to *Acinetobacter* spp. and other Gram-negative rods. Other studies have also reported the growth of GPB and GNB in preservative-free multi-dose eye drops used by patients and HCP. Contaminants of the eye drops were CoNS, *S. aureus*, *Bacillus* spp., *Serratia* spp., *Klebsiella oxytoca*, *Enterobacter cloacae*, and *α viridans streptococci.* [37]. Kim (2008) [47] investigated the microbial contamination of preservative-free artificial tears in reclosable containers and found CoNS and *Acinetobacter* spp. [47].

Different studies found bacterial contamination with GPB and GNB but also demonstrated contamination with fungi. Schein (1992) [41] tested different types of preserved ophthalmic solutions handled by patients. The isolated microorganisms were GPB such as CoNS, *diphtheroids*, *S. aureus*, *Propionibacterium (Cutibacterium)* spp., and *Streptococcus* spp., as well as fungi found in the bottle cap. GNB included *Pseudomonas* spp., *Proteus* spp., *Klebsiella* spp., *Serratia* spp., *Enterobacter* spp., *Citrobacter* spp., *Acinetobacter* spp., *Haemophilus parainfluenzae*, and non-enteric Gram-negative rods cultured from multiple sites [41]. Clarck (1997) [19] found *Pseudomonas putida*, *S. epidermis*, *Micrococcus luteus*, and yeasts in diagnostic ophthalmic solutions in primary care settings [19]. The residual content and the tip were both contaminated. The author highlighted the presence of dried microscopic residues on the thread of the bottle tip that constituted a likely risk of contamination to patients [19]. Livingstone (1998) [30] examined the extended use of eye drops in the medical and surgical hospital wards. The identified microorganisms from the residual contents of eye drops were CoNS, *Micrococcus* spp., *Bacillus* spp., *P. mirabilis*, *Serratia liquefaciens*, and less frequently, different fungi such as *Cladosporium spp*., *Penicillium* spp., and unidentified yeasts. The mean counts of these microorganisms were low, and the author considered the microbes part of the skin flora. Gram-positive spore-bearing bacilli and fungal spores are contaminants present in the air and not a clinically significant threat to the patient [30]. These findings are consistent with a similar published study [84]. Fazeli (2004) [23] studied multi-user eye drops collected from an outpatient setting and found high contamination of the cap and residual content with *S. epidermidis*, *S. aureus*, *Micrococcus* spp., *Corynebacterium* spp., *Bacillus* spp., and fungi, including *Aspergillus niger*, *Cladosporium* spp., *Aspergillus fumigatus*, *Aspergillus flavus*, other *Aspergilli*, *Mucor* spp., and *Geotrichum* spp. These microorganisms are part of the human flora or the environment, such as airborne Gram-positive spore-bearing bacilli and fungal spores. There was no GNB detection. Contamination of the cap and residual content was higher compared with the tip. Previous studies had similar results [38]. Thanathanee (2013) [14] showed that preservative-free 100% autologous serum eye drops prepared on-site in a hospital were contaminated with GPB, such as CoNS, *Bacillus* spp., and *Corynebacterium* spp., in addition to fungi, including *Aspergillus* spp. and *Fonsecaea* spp. [14]. These findings were attributable to uncapped bottles post-use, and the authors considered that the eye drops were possibly left open [23]. Feghhi (2008) [48] tested different eye drops in a hospital ophthalmology department and found that the isolated microorganisms were either fungi or GPB from human flora and airborne Gram-positive spore-bearing bacilli. The GPB included *Bacillus cereus*, *Bacillus subtilis*, *diphtheroids* spp., *Nocardia* spp., and CoNS. The fungal isolates were environmental saprophytes, including yeasts such as *Candida albicans* and *Rhodotorula rubra.* The most common molds were *A. flavus*, *Penicillium* spp., and *Cladosporium* spp., in addition to *A. niger*, *Gliocladium* spp., *Acremonium* spp., and *Alternaria* spp. These findings were attributable to seasonal airborne fungi and environmental conditions such as temperature and humidity. In this study, the samples were collected during the springtime when the weather was favorable for fungal air dispersion. Razooki (2011) [39] found that most of the bacteria identified belonged to the normal commensal flora of the conjunctiva or the skin, such as *S. aureus*, *S. epidermidis Micrococcus* spp., and *Neisseria catarrhalis* [39]. Other isolated microorganisms were fungi, including *C. albicans* [39]. Bachewar (2018) [15] studied the contents of multi-user eye drops in an outpatient hospital setting. Isolates from the caps and residual content were *E. coli*, *P. aeruginosa*, *S. aureus*, *B.subtilis*, and C. *albicans*.

Teuchner (2015) [46] compared the microbial contamination of ophthalmic formulations used by patients versus HCP to treat glaucoma. The drops and the residual content showed contamination by mainly GNB and bacterial spores [46]. The microbial contamination of glaucoma eye drops used by patients was predominantly isolated from the dropper tip of the container. Human opportunistic microorganisms and pathogens were found mainly in samples collected from homes, such as *P. aeruginosa*, *S. marcescens*, *Acinetobacter lwoffii*, *S. maltophilia*, and *S. aureus*. Other microorganisms found in human flora and the environment included *Staphylococcus* spp., *M. luteus*, *Bacillus* spp., *Streptococcus* spp., *Corynebacterium* spp., *Neisseria* spp., *Rothia dentocariosa*, *Aerococcus viridans*, *Moraxella osloensis*, *Kocuria rosea*, *Arthrobacter* spp., *Pantoea agglomerans*, *Streptomyces violaceoruber*, *Brevibacterium casei*, *Cellulosimicrobium cellulans*, and filamentous *Aspergillus* spp.

Kyei (2019) [28] studied diagnostic eye drops used in an ophthalmology clinic and obtained heavy growth of microbial contaminants. The isolated microorganisms were bacteria found in the human flora of the eye, skin, nasopharynx, and gastrointestinal tract, in addition to fungi. The airborne microbial contaminants were attributable to the geographical area, the environmental conditions, and hygiene-related factors [28]. Kyei (2019) [29] also evaluated topical therapeutic eye drops and found similar results from diagnostic eye drops, in addition to *Enterobacter* spp. and *Alternaria* spp. [29].

Chantra (2022) [17] tested preservative-free eye drops collected from homes and in a hospital and found contamination with GPB, GNB, and fungi [17]. Molds, such as *Aspergillus* and *Fusarium*, were the most frequently isolated contaminants, followed by GPB. The microbial isolates were *Staphylococcus* spp., *M. luteus*, *B. cereus*, *Corynebacterium* spp., *P. aeruginosa*, *S. maltophilia*, *Neisseria* spp., *E. coli*, *Kocuria rhizophila*, *Arthrobacter*, *E. coli*, *Brevibacterium casei*, *Exiguobacterium* spp., and fungi, including yeasts, *Candida* spp., yeast-like basidiomycetes, and *Trichosporon asahii.*

The variation in the fungi detected depends on the culture methods used, especially the selected culture media and the incubation conditions (temperature and delay). At the same time, their detection is related to environmental contamination and the long-term use of eye drops.

Even if reported microbial contamination is low, some studies found contamination with antibiotic-resistant strains (natural or acquired resistance) that represented a risk of ocular injury to patients [50]. Tsegaw (2017) [50] reported contamination of the dropper tip with MRSA, while there was none in the residual content. Tamrat (2019) [43] found multiple drug-resistant bacteria among GNB, such as *Klebsiella* spp. and *Pseudomonas* spp. In this study, the dropper tip was more contaminated than the drops. Kyei (2019) [29] showed that most isolated bacterial contaminants were resistant to antibiotics, including *Pseudomonas* spp. as the most resistant cultured strain. Figuêiredo (2018) [24] found that GNB isolated from the drops of ophthalmic solutions originating from hospital settings were resistant to conventional antibiotic therapy. Antimicrobial-resistant microorganisms represent a challenge in treating eye infections, predominantly in patients with compromised ocular conditions. Treatments are mainly probabilistic, without sampling or identification of the microorganism involved. As a result, the risk of ocular injury due to infection with antibiotic-resistant strains, such as Methicillin-resistant *Staphylococcus aureus* (MRSA) and *P. aeruginosa*, indicates that the dropper tip is a serious source of eye drop contamination (Table 4).

### 3.5. Factors Associated with Microbial Contamination

#### 3.5.1. Single-User versus Multi-User

Numerous studies have reported inconsistent findings across settings (outpatient, ward, operating theater, private clinic, home) and users (patient, healthcare professionals).

Studies have reported higher microbial contamination of preserved eye drops when handled by multiple users compared with a single user [42,50], while others have demonstrated opposite findings [34,43,46].

The differences between microbial contamination rates handled by single users versus multiple users were only significant in the study conducted by Teuchner (2015) [46].

Two studies showed that preservative-free hospital-prepared eye drops collected from patients had higher microbial contamination rates than those used by HCP [17,37]. These differences were significant in only one study by Rahman (2006) [37].

Microbial cultures consistently showed that the dropper tip was the most frequently contaminated site [17,34,43,46,50].

#### 3.5.2. Inpatient versus Outpatient Settings

The microbial contamination rates of in-use topical ophthalmic solutions in the OR are lower than in the ward and outpatient departments [21,30,43,46].

Teuchner (2015) [46] found the differences statistically significant. Nenwitch (2007) [34] showed no significant differences in the contamination rates of eye drops handled by multiple users between the ward and outpatient settings and found no contamination of eye drops in the OR. The sample size in this study was limited [21,34].

Mason (2005) [57] tested the microbial contamination of topical ophthalmic antibiotic solutions (moxifloxacin) used by patients at home versus multiple uses by HCPs. A low contamination rate of the thread of eye drop bottles with dried residues was found in the home-used eye drops. Other studies were conducted in extended-term care facilities [22], primary eye care clinics [19,29], or home-used ophthalmic solutions and showed variable contamination rates [28].

The studies that compared the microbial contamination of eye drops in the OR considered a limited duration of use when compared to other hospital settings with extended in-use time frames [21,46]. More standardization of the duration of use is necessary to evaluate the microbial contamination rates and CFU counts of ophthalmic medications taken from OR samples and compare them with other settings.

#### 3.5.3. Duration of Use

The duration of eye drop use did not consistently influence the microbial contamination rates in many published studies [15,22,24,26,30,46,48,85].

Feghhi (2008) [48] did not document an association between the duration of use and the microbial contamination rate of eye drops collected from inpatient hospital settings between day one and day seven [48]. Fazeli (2004) [23] conducted a similar study on eye drops taken from outpatient settings and found a significant difference (between day one and day seven) for the same period of use.

Livingstone (1998) [30] showed that stretching the recommended period of use of ophthalmic solutions in inpatient settings from seven days (6.1%) to fourteen days (9.4%) did not lead to a significant increase in the rate of microbial contamination and did not represent a clinical threat to patients [30]. Livingstone (1998) [30] recommended extended use to generate healthcare savings [30]. Hanssens (2018) [85] demonstrated that diagnostic eye drops recommended for use for up to 28 days (manufacturer) could be safely used in a controlled clinical context for up to 7 months [85]. The documented microbial contamination rate was 2.69% after 6 months [85]. Mehr-un-Nisa (2019) [33] showed that multi-dose anesthetic eye drops could be used for one month after opening without a risk of infection [33]. In this study, no growth was detected in the tested eye drops. The author stated that they took cultures from the content of the bottle only, and the sample size was limited; they recommended conducting future research on a large scale to confirm their findings [33].

Chua (2021) [18] had different findings and considered that the increase in the contamination rate of eye drops collected from outpatient settings on day fourteen was affected by the tropical weather compared with the temperate country in the previous study [18]. In all cases, the eye drops were from multi-user settings, and a positive association occurred in outpatient settings.

Numerous other studies have demonstrated an association between the microbial contamination rate and the duration of use in hospital settings [2,3,6,8,34]. Tsegaw (2017) [50] found that bacterial contamination of eye drops increased from 3.2% (less than seven days) to 24.3% (more than seven days) and demonstrated that frequent use occurring for an extended time led to a higher risk of microbial contamination [50]. Teuchner (2015) [46] compared the microbial contamination of in-use eye drops after one week in the OR and four weeks in the medical ward, outpatient units, and home. The results showed that a shorter duration of use in the OR (one week) was associated with a lower contamination rate [46]. Tamrat (2019) [43] conducted a study on eye drops used for two weeks and showed an increase in the contamination rate with duration and lower frequency of use (less than four times per day). Geyer (1995) [49] tested glaucoma eye drops used by patients and showed an increase from 19% after less than eight weeks of use to 40%. In these studies, the tested eye drop bottles contained a preservative expected to prevent microbial contamination for the recommended duration of use. Multiple microorganisms showed low sensitivity to preservatives, such as GNB to quaternary ammonium [46], and a lack of preservative effect on bacterial spores. At the same time, the low concentration of preservatives may limit their activity against microbial contaminants. Teuchner (2015) [46] underlined that balanced efficacy and toxicity of preservatives are needed to protect the ocular surface and prevent microbial contamination. In all cases, preservatives can only contribute to the limitation of eye drop content contamination, but they do not have a role in preventing dropper tip and cap contamination due to limited contact time. This can leave these eye drop sites exposed to microbial contamination from humans and the environment, as well as potential growth on these surfaces.

The discrepancy in the role of the duration of use in increasing the risk of eye drop contamination and, subsequently, patient infection may be related to the limitations of these studies, which include, limited sample sizes of the tested eye drops; sites of eye drop cultures often limited to the drops but not the dropper tip, cap and residual content; and the settings (controlled or not). The available studies did not offer consistent evidence to reach a consensus about the optimum duration of use depending on the type of eye drops used.

#### 3.5.4. Frequency of Use

Eye drops used multiple times per day are assumed to be at increased risk of microbial contamination. However, a limited number of studies have evaluated the significance of this association. Chua (2021) [18] considered that the higher contamination rate of local anesthetic eye drops compared with mydriatics may be explained by the frequency of use. Chantra (2022) [17] found that the number of eye drops per person was significantly associated with a higher rate of microbial contamination (*p* < 0.022). Other studies confirm these findings [24,34].

Tamrat (2019) [43] obtained different results and showed a higher contamination rate in eye drops instilled less than four times per day compared with more frequent use. The authors postulated that medications applied less frequently may be in use for an extended period of time, leading to an increased risk of contamination [43].

#### 3.5.5. Handling Techniques

The risk of microbial contamination is attributed to poor handling techniques [23,32,44], even if handled by an HCP [18,41,47,50]. The improper use of ophthalmic solutions is a documented risk factor for microbial contamination [28,36]. During instillation, the dropper tip may inadvertently touch the fingertips [19,47] or come in contact with the facial skin [7], eyelids, ciliary eyelashes, ocular surfaces [7], nose, and surprisingly, even the mouth [19]. The same risk occurs during the formation and application of the drops. Older-age patients are more prone to this risk during self-administration [24,32,47,49] owing to poor vision and dexterity [17,37]. In these cases, a patient may need help to instill the eye drops [22,43]. Studies showed that trained persons had a lower contamination rate of ophthalmic solutions [26,86]. Chua (2021) [18] considered that the variability in the contamination rate between tested diagnostic eye drops was due to differences in the handling technique upon medication administration by a physician under a slit lamp or by a healthcare worker at a clinic in the waiting room area. Following strict hygiene measures is recommended, even if their role in preventing microbial contamination is not clearly demonstrated, as noted in tested eye drops collected from the OR [21,30,46]. Bachewar (2018) [15] showed that contamination risk was still prevalent, even if a patient followed the instructions of use. Storage conditions are an additional risk factor. Figuêiredo (2018) [24] reported that, when patients were asked if they were sure to follow the instructions, they kept their eye drops stored in their room, living room, bathroom cabinet, refrigerator [24], or purse when frequent ocular drug administration was needed per day.

Dacosta (2020) [87] discussed the role of the instillation angle in microbial cross-contamination of multi-dose eye drops and found that increasing the angle of instillation of the ophthalmic solution from 45° to 90° may reduce the contamination rate.

Most studies have cited many factors associated with microbial contamination and have considered different study designs, microbiological cultures, and analyses, in addition to different settings, lengths of use, formulations, and targeted populations. As a result, the comparison of findings is difficult due to multiple other inconsistencies, which highlights the need to address these gaps in future research.

## 4. Discussion

Our review showed that the microbial contamination of in-use eye drops ranged between 2 to 94% [15,17,18,19,21,22,23,24,26,27,28,29,30,34,35,36,37,39,41,42,43,44,46,47,48,49,50,56,57]. Not all studies examined the microbial contamination of the dropper tip and cap. Fifteen out of twenty-eight studies that took cultures from all the sites, including the dropper tip, cap, residual content, and drops, showed that the microbial contamination rates of the dropper tip and cap varied from 7.7 to 100% of the total contaminated samples [17,18,19,21,23,34,39,41,43,46,48,49,50,56].

The microbial contamination of the eye drops and, especially, of the dropper tip and cap are of high relevance because:(1)A dropper tip offers a wide surface that is exposed to human and environmental microorganisms;(2)A contaminated dropper tip can come in contact with the ocular surface, eyelids, and eyelashes during drug self-administration or the instillation of eye drops by another person, as predominantly documented in the elderly;(3)A contaminated dropper tip and cap can lead to the contamination of eye drop content, as shown previously;(4)Studies showed contamination of the dropper tip with antibiotic-resistant bacteria;(5)Microbial contamination was demonstrated even when the eye drops were handled by HCP or in the OR;(6)The preservatives did not have sufficient contact time with the dropper tip, cap, or thread to exert their effect;(7)The dropper tip is a documented source of ocular infections and, subsequently, eye injuries, such as keratitis and corneal ulcers [45,52,56].

Numerous studies have reported the predominant microbial contamination of in-use eye drops with commensal human and environmental flora. The commensal flora is nonharmful to humans and part of the ocular protective mechanisms [16,17,21,29,34,36,88]. Under certain conditions, such as post-ophthalmic surgery, some commensal bacteria may lead to severe ocular injuries, such as endophthalmitis caused by CoNs bacteria [17,88]. Other cultured microorganisms detected at very low concentration were pathogenic, including resistant bacteria and fungi [21,37,41,43,46,49]. In rare cases, human pathogens usually found in the gastrointestinal tract and nasopharynx were cultured from different sites of the ophthalmic products [46]. These pathogens were mainly Enterobacteriaceae, such as *E. coli*, *Klebsiella* spp., *Serratia* spp., *Proteus* spp., *Salmonella* spp., and *Shigella* spp. [46]. Depending on the type of bacteria, these results showed that they may originate from fecal contamination, probably by hands or contact with the nasopharyngeal area and potentially linked to inadvertent touching of the tip of the eye drops during instillation or during manipulation of the cap to open the bottle. *Serratia marcescens*, a frequently isolated bacterium from contaminated eye drops, can cause keratitis [45], corneal ulcers [53], and endophthalmitis [89]. Other commonly cultured bacteria, *Pseudomonas* spp., show intrinsic and acquired antibiotic resistance [21,23,24,29,43] and are considered environmental contaminants with a high ability to colonize and then infect patients as an opportunistic pathogen with nosocomial significance. The bacteria belonging to this genus have the potential to cross an uncompromised corneal epithelium and lead to corneal damage [28,90,91]. The spore-forming *Bacillus* spp. is extensively cultured from tested eye drops because preservatives fail to destroy spores and sometimes prevent secondary vegetative growth [46].

Contaminated eye drops are a risk for microbial ocular injury [32,34,48,56,89,90,91,92,93,94,95,96,97,98,99]. Keratitis is a leading cause of corneal opacity, perforation, and potentially, visual loss and blindness [92]. Endophthalmitis, a rare, acute invasive condition attributable to GNB such as *Klebsiella* spp., *E coli*, and GPB such as *Staphylococcus* spp., *Streptococcus* spp., in addition to fungi, can lead to irreversible blindness [93,94]. Post-surgical endophthalmitis accounts for the majority of endophthalmitis documented in multiple countries [100,101].

Unit-dose formulations potentially offer risk-free ocular drug delivery, are much more expensive than multi-dose eye drops, and can generate a large amount of plastic [25,47]. The dropper tip touches the fingertips when twisting off the dropper tip to open the unit-dose container. Microbial contamination of these dosage forms is documented in reclosable containers, while research in this domain is still scarce [47]. Although recommended for one-time use, people may tend to keep the excess content for more frequent use [47]. Somner (2010) [25] examined the financial and environmental impacts of reducing to zero the risk of dropper tip contamination by using disposable minims only once. The estimated costs ranged between GBP 2.75 and 4.6 million per year, while the expected number of generated waste was between 6.85 and 11.42 tons for paper waste and 12.69 to 21.15 tons for plastic waste [25]. The poor effect of preservatives, predominantly BAK, has been highlighted in multiple studies [15,18,19,21,23,24,29,30,34,43,46,48,49,50,52,60], predominantly without addressing their antimicrobial efficacy [46]. A spray format may be an option for meeting preservative-free needs, but this design does not cover all therapeutic and diagnostic medications [56]. The risk of microbial contamination via spray splash is not yet determined. Other alternatives include innovative technology dispensers, now available in the market [82,83]. These are high-security containers based on a built-in filter or valve technology for a mechanical, integrated, airless application system that ensures the one-way flow of the content to protect and maintain the long-term sterility of the contents of eye drops. These primary packaging methods are not marketed yet in different formulations intended to diagnose or treat a wide array of diseases, may be expensive, and may not be covered by third-party payers [82,83].

An innovative green solution is also available. This technology consists of a self-decontaminating dropper tips and caps that are mercury- and metal-free. This technology incorporates mineral microspheres into any plastic resins (tip and cap of a multi-dose container) to exert a disruptive antimicrobial effect without causing an allergic reaction or patient harm [81].

Studies have extensively mentioned the formulations of eye drops, including the active ingredients and preservatives (when applicable), as a factor associated with the microbial contamination of eye drops but are not conclusive about their effect. Only one study by Porges (2004) discussed the susceptibility of latanoprost to bacterial contamination [36]. More research is needed to show the susceptibility and related factors of formulations to microbial contamination. Other factors include the duration of use [16,23,37,39,50], the number of drops per day [20,23,26,34,46], the frequency of use [43], multiple users of the same container [18], handling by older-age users [24,49], the setting (including outpatient and inpatient, even in the operating theater) [15,17,21,46], patient compliance with instructions issued, instillation angle [22,23,87], bottle geometry [22,23,87], tip shape [18,23], color [102], the physicochemical properties of the eye drops [40], doubtful storage conditions [47], personal hygiene, and microorganisms found under the nails (such as fungi) [17].

## 5. Recommendations

Our review found that published studies may have limitations that can influence the reliability of the findings. The sample size, the sampled sites (not only eye drop content), the methods of specimen collection, the transfer of samples, the culture technique, the uses of a variety of media, the incubation conditions, and the microbiological assays varied across different studies. Participants may exhibit a more careful behavior contributing to bias when they are aware of ongoing research and if eye drops are labeled, mainly in settings with strict regulations, such as hospitals. The dropper tip, cap, residual content, and thread were not always cultured, and the CFU counts in each tested eye drop site were not always mentioned. The method of microbiological analysis did not always cover all possible contaminants, such as microaerophilic contaminants, spores, anaerobes, and fungi. There were no or only a few studies discussing the link between species identification from the different parts of the product (dropper tip, cap, drops, residual content), quantification of the contamination level, and the infectious risk. Despite this, the microbiological results of microorganisms and pathogens involved in ophthalmic product contamination and eye infection were usually consistent. More studies are needed to investigate the association between eye injury and microbial contamination because research in this domain is scarce but highly relevant to advising health policy.

Our review highlighted the need for a high-quality design study conducted in multiple settings to assess the microbial contamination rates of eye drops. The tested eye drops must be the same medication and from the same manufacturer, containing a similar preservative concentration (if applicable). The expected results could inform the influence of eye drop formulations.

There is a need to implement standardized protocols to allow generalizability and comparisons of findings between different countries while accounting for the environmental, socio-cultural, and socio-economic differences specific to each country and limited resource settings. More focused research is needed to determine the contribution of preservative-free unit-dose containers and innovative new technologies to the prevention of microbial contamination under real-life scenarios.

## 6. Conclusions

The dropper tips and the caps of in-use eye drops are sources of microbial contamination and a risk of infection and ocular injury in susceptible patients. Evidence has shown that the handling of eye drops by patients, caregivers, and healthcare providers is a risk for dropper tip contamination. The other contributory factors of microbial contamination were inconsistent across the literature to reach a consensus of use and determine the true magnitude and the clinical impact of contamination for these eye drop sites. Many questions remain unanswered, such as the impact of the frequency of use, extended use beyond the recommended period, the re-use of unit-dose formulations (including generated plastic waste), and the associated costs. New technologies offer a promising potential for securing the long-term sterility of in-use eye drops and could also benefit from a demonstration of effectiveness in the framework of a well-defined study protocol.

## Figures and Tables

**Figure 1 pharmaceutics-14-02176-f001:**
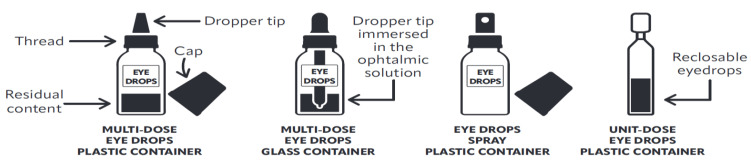
Frequent types of in-use eye drops.

**Figure 2 pharmaceutics-14-02176-f002:**
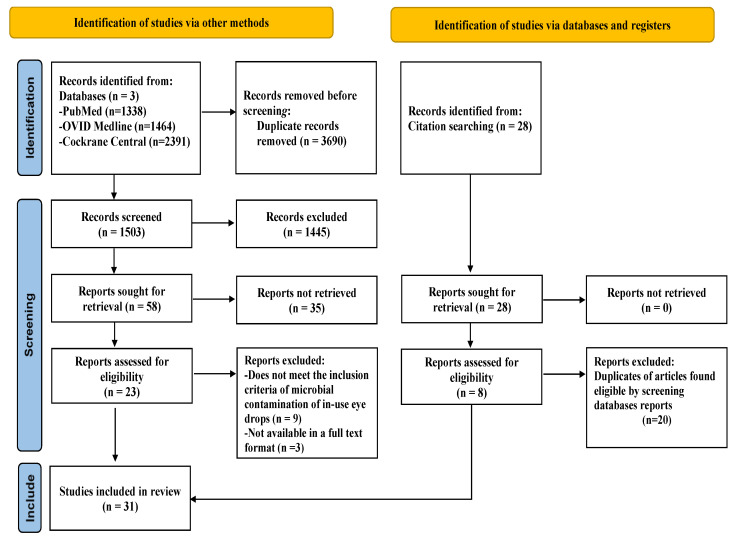
PRISMA flow diagram of the search.

**Table 1 pharmaceutics-14-02176-t001:** List of the 31 studies investigating the microbial contamination of eye drops and their principal features.

Authors	Year of Publication	Country	Study Design	Aim of the Study	Setting
Chantra et al. [17]	2022	Thailand	Cross-sectional study	The aim of this study was to assess the incidence of microbial contamination in preservative-free hospital-prepared anti-infective eye drops and investigate factors that contributed to contamination.	Hospital
Chua et al. [18]	2021	Malaysia	Cross-sectional study	To determine the prevalence of microbial contamination in multi-user preserved ophthalmic drops in an ophthalmology outpatient clinic to compare the rates of contamination between the dropper tip and the residual contents in the bottle and to identify the contaminating organisms.	Hospital
Daehn et al. [21]	2021	Germany		To address the potential contamination of multi-dose ophthalmic solutions in the operating theater and the underlying risk of infection by examining the microbiological load on the tips of dispenser bottles.	Hospital
Kyei et al. [27]	2019	Ghana		To investigate the possible microbial contamination of fluorescein sodium dye solutions used in eye clinics in Ghana.	Eye care clinics
Kyei et al. [28]	2019	Ghana	Cross-sectional study	To determine the microbial contaminants and their clinical importance in topical diagnostic ophthalmic medications in eye clinics in Ghana.	Eye care clinics
Kyei et al. [29]	2019	Ghana	Clinical experiment	To evaluate the microbial contamination of in-use therapeutic ophthalmic medications in the Cape Coast metropolis.	Home
Nisar et al. [35]	2019	Pakistan		To investigate the bacterial contamination of eye drops dispensed for multi-dose purpose.	
Tamrat et al. [43]	2019	Ethiopia	Cross-sectional study	To determine the magnitude of contamination and pattern of antimicrobial resistance of in-use ophthalmic solutions.	Hospital
Bachewar et al. [15]	2018	India	Prospective observational study	To determine the magnitude and pattern of microbial contamination rates in multi-dose used eye drop containers and residual medicine in presence or absence of preservatives.	Hospital
Figuêiredo et al. [24]	2018	Brazil	Cross-sectional study	To evaluate contamination in topical medication eye drops of patients from the glaucoma ambulatory of a university hospital and use a questionnaire to analyze the storage and method of instillation of the eye drops collected.	Hospital
Tsegaw et al. [50]	2017	Ethiopia	Cross-sectional study	To assess the magnitude and pattern of bacterial contamination of multi-dose ophthalmic medications and investigate the drug susceptibility pattern of the isolates in the Department of Ophthalmology at Gondar University Teaching Hospital.	Hospital
Teuchner et al. [46]	2015	Austria		To compare the percentage of contamination of multi-use eye drops applied by glaucoma patients at home and by the medical personnel in the outpatient department, ward, and operating room of a Department of Ophthalmology and to test the influence of sampling from the eye drop tips, drops, and residual fluid inside the bottle.	Hospital
Thanathanee et al. [14]	2013	Thailand	Prospective descriptive study	To evaluate the sterility and safety of 100% nonpreserved, autologous, serum eye drop treatment in patients with ocular surface diseases.	Hospital
López-García et al. [12]	2012	Spain	Prospective, consecutive, comparative, and randomized study	To assess the effect of the use of containers with adapted sterilizing filters on the contamination of autologous serum eye drops.	Home
Razooki et al. [39]	2011	Iraq		To determine the magnitude and pattern of microbial contamination of eye drops in outpatients at the department of ophthalmology.	Hospital
Somner et al. [25]	2010	UK		To quantify the financial and waste implications of reducing this risk to zero by using disposable droppers only once.	Eye care clinics
Feghhi et al. [48]	2008	Iran		To investigate the incidence of fungal and bacterial contaminations of in-use eye drop products in the teaching department of ophthalmology.	Hospital
Kim et al. [47]	2008	Republic of Korea	Prospective, non-masked, randomized trial	To evaluate microbial contamination of multiple-use preservative-free artificial tears packed in reclosable containers after daily use.	Home
Nentwich et al. [34]	2007	Kenya	Cross-sectional study	To determine the magnitude and pattern of microbial contamination (bacterial and fungal) of multi-dose ocular solutions.	Hospital
Jokl et al. [22]	2007	USA		To assess the frequency of contamination of ophthalmic solutions in a long-term care facility and to describe the characteristics of contaminated solutions.	Long-term care facility
Rahman et al. [37]	2006	UK		To investigate the incidence of microbial contamination in preservative-free drops dispensed from multi-use containers.	Hospital
Mason et al. [57]	2005	USA	Prospective, non-masked, non-randomized trial	To determine the contamination rate of topical moxifloxacin 0.5% (Vigamox) after clinical use for preoperative and postoperative prophylaxis for cataract surgery.	Hospital
Porges et al. [36]	2004	Israel	Cross-sectional study	To evaluate the sterility of topical glaucoma medications among chronic glaucoma medication users in the community.	Community
Fazeli et al. [23]	2004	Iran		To assess the validity of an increased in-use period for preserved eye drops opened in a hospital outpatient department.	Hospital
Lagnado et al. [13]	2004	UK		To establish if contamination of 20% autologous serum drops prepared under sterile conditions occurred over a 24 h period of one to two hourly use in a hospital inpatient setting.	Hospital
Livingstone et al. [30]	1998	UK	Comparative study	To compare the microbial contamination of eye drop residues used by inpatients for both 7 and 14 days in order to assess the validity of an increased in use period for preserved eye drops issued to hospital inpatients.	Hospital
Clarck et al. [19]	1997	USA		To investigate the possible contamination of a representative sample of diagnostic pharmaceutical agents and irrigating solutions in small office practices.	Eye care clinics
Donzis [56]	1997	USA	Case report	To report a complication of aerosol saline use in a contact lens wearer.	Home
Geyer et al. [49]	1995	USA	Comparative study	To estimate the frequency of contamination of topical antiglaucoma medications used by asymptomatic patients.	Hospital
Schein et al. [41]	1992	USA	Comparative study	To estimate the frequency of medication contamination and to test the hypothesis that contaminated medications were associated with conjunctival colonization with the same organism.	Hospital
Stevens and Matheson [42]	1992	UK		To assess whether short-stay patients having routine surgery who used postoperative eye drops had contamination of these drops on leaving hospital.	Hospital

Abbreviations: UK, United Kingdom; USA, United States of America.

**Table 2 pharmaceutics-14-02176-t002:** Microbial contamination of in-use eye drops.

Settings						Type of Product	Sample Size	Contains a Preservative		Rate of Microbial Contamination	Site of Contamination					Duration of Use	Ref.
	Inpatient Ward	Surgical Theatre	Outpatient Clinics	Single User Setting	Multi-User Setting			Yes	No		Dropper Tip	Drops	Residual Content	Cap	Dried Residue Thread		
Community				X		Multi-dose	156	X		29%		X	X	X		>2 month	[41]
Hospital	X		X	X		Multi-dose	216	X		2%		X				≤72 h	[42]
Community				X		Multi-dose	194	X		28%	20%	8%				>3 month	[49]
Eye care clinics			X	X	X	Multi-dose	60	X		12%	X	X		X	X		[19]
Home				X		Multi-dose	1		X		X	X					[56]
Hospital	X	X			X	Multi-dose	31 (D7); 295 (D14)	X		6% (D7) vs 9% (D14)			X			1-2 weeks	[30]
Community				X		Multi-dose				13%						1-12 weeks	[36]
Hospital			X		X	Multi-dose	200	X		44% (D1) vs 70% (D7)			50%	32%		1-7 days	[23]
Hospital				X		Multi-dose	134		X	9.70%		X				Day 0 and day 1 for a minimum of four consecutive days and a maximum of 14 days	[13]
Hospital		X		X	X	Multi-dose	61		X	2.00%					X	Pre-OP:2.2 days; Post-OP: 7.2 days	[57]
Hospital	X		X	X	X	Multi-dose	95		X	8%		X				D3 (Inpatients); D7 (outpatients)	[37]
Hospital	X	X	X	X	X	Multi-dose	101	X		6%	5%	0%				2 weeks	[34]
Long-term care facility	X			X		Multi-dose	123	X		8%		X				1 week	[22]
Hospital	X				X	Multi-dose	287	X		18%	41%		13%	46%		Day 1, 2, 3, 4 and 7	[48]
Home				X		Multi-dose and unit dose	207	X (control)	X	2%		X				10 hr	[47]
Eye care clinics				X		Unit dose	100		X	5%	5%					Instant use	[25]
Hospital			X		X	Multi-dose	54	X		15%	9%	4%				Average2 weeks	[39]
Home					X	Multi-dose	176 *		X	2% (container with filter) and 29% (conventional containers)		X				Conventional containers: 1 week; Containers with adapted filters: days 1, 14, 21, 28	[12]
Hospital					X	Multi-dose	147		X	6%		X				Daily for 1 week starting day 0	[14]
Hospital	X	X	X	X	X	Multi-dose	400	X		17%	20% **	8% **	5% **			1 week (OR) vs. 4 weeks other settings	[46]
											1% ***	11% ***	7% ***				
Hospital				X	X	Multi-dose	100	X		11%	11%		0%			≥1 week	[50]
Hospital			X		X	Multi-dose	55	X		25%	X		X	X		1–8 weeks	[15]
Eye care clinics			X	X		Multi-dose	55	X		9%		X				>1 month	[24]
Home				X		Multi-dose	21	X		100%		X					[27]
Eye care clinics				X		Mulit-dose	113	X		96%						2–8 weeks	[28]
Eye care clinics				X		Multi-dose	100			65%			X			≤2 weeks	[29]
Hospital			X	X		Multi-dose	106			23%							[35]
Hospital	X	X	X	X	X	Multi-dose	70			73%	61%		4%			Average 12 weeks	[43]
Hospital		X			X	Multi-dose	245			2%	X					≤5 days	[21]
Hospital	X		X	X	X	Multi-dose	140	X		30%	50%		33%			Day 14 and day 30	[18]
Hospital			X		X	Multi-dose	295		X	24%	49%		27%			>2 days	[17]

The rates of contamination of eye drops were calculated out of the total tested eye drop samples. The microbial contamination rates of the dropper tips, caps, drops, and residual contents were calculated out of the total contaminated eye drop samples. X: yes; * 48 containers with adapted filters and 128 conventional containers; ** patient; *** healthcare professional. Abbreviations: HCP, healthcare professional; hr, hour; OP, operative; OR, operating room.

**Table 3 pharmaceutics-14-02176-t003:** Types of tested eye drop medications.

Authors	Tested Eye Drops Medications																										Use of Preservatives																
	Mydriatics	Glaucoma Medications	Miotics	Anesthetics	Steroids	Antimicrobials	Antimicrobials/Glucocorticoids	Antibiotics	Antifungals	Lubricants	Antiinflammatory Agents	NSAIDs	Antihistamines	Others	Acetylcysteine	Aerosol Saline Spray	Autologous Serum	Artificial Tears	Cyclosporine	Fluorescein Solutions	Hypromellose	Idoxurudine 0.1%	Irrigating Solutions	Methylcellulose 2%	Combinations	Atropine Sulfate 1%, Phenylephrine HCl 2.5%, Cyclopentolate HCl 1% and Tropicamide 1%	Preservative Free	Preservative	Benzalconiumchloride 0.004%	Benzalkonium 0.005%	Benzalconiumchloride 0.01%	Benzalconiumchloride 0.014%	Benzalconiumchloride 0.02%	Benzalconiumchloride 0.03%	Benzalkonium 0.05%	Chlorbutol IP 0.5%	Cetrimonbromide	Hydroxybenzoate	Phenylmercuric Nitrate IP 0.001%	Phenylmercuric Nitrate IP 0.002%	Sodium Perborate	Thiomersal 0.005%	Thiomersal 0.001%
Chantra et al. [17]																																											
Chua et al. [18]																																											
Daehn et al. [21]																																											
Tamrat et al. [43]																																											
Nisar et al. [35]																																											
Kyei et al. [29]																																											
Kyei et al. [28]																																											
Kyei et al. [27]																																											
Figuêiredo et al. [14]																																											
Bachewar et al. [15]																																											
Tsegaw et al. [50]																																											
Teuchner et al. [46]																																											
Teuchner et al. [46]																																											
Thanathanee et al. [14]																																											
Lopez-garcia et al. [12]																																											
Razooki et al. [39]																																											
Somner et al. [25]																																											
Kim et al. [47]																																											
Feghhi et al. [48]																																											
Jokl et al. [22]																																											
Nentwich et al. [34]																																											
Rahman et al. [37]																																											
Mason et al. [57]																																											
Lagnado et al [13]																																											
Fazeli et al. [23]																																											
Porges et al. [36]																																											
Livingstone et al. [30]																																											
Clarck et al. [19]																																											
Donzis [56]																																											
Geyer et al. [49]																																											
Stevens and Matheson [42]																																											
Schein et al. [41]																																											

**Table 4 pharmaceutics-14-02176-t004:** Isolated microorganisms from tested eye drops.

Eye Drops		Microbial Contamination Sites					Gram-Positive Bacteria																Gram-Negative Bacteria																						Fungi																	Ref.
Preservative Free	Preservative	Dropper Tip	Drops	Residual Content	Cap Contaminated	Dried Residue Thread	*Gram-positive cocci*	*Aerococcus viridans*	*Arthrobacter* spp.	*Bacillus* spp.	*Brevibacterium casei*	*Clostridium perfringens*	*Corynebacterium* spp.	*Kocuria* spp.	*Micrococcus* spp.	*Propionibacterium acnes*	*Rothia dentocariosa*	*Staphylococcus coagulase* negative	*Staphylococcus epidermidis*	*Staphylococcus aureus*	*Staphylococcus* spp.	*Streptococcus* spp.	Gram Negative Rods	*Acinetobacter* spp.	*Bordetella* spp.	*Citrobacter* spp.	*Escherichia coli*	*Edwardsiella*	*Enterobacter* spp.	*Haemophilus* spp.	*Flavobacterium* spp.	*Klebsiella* spp.	*Moraxella* spp.	*Morganella morganii*	*Neisseria* spp.	*Nocardia* spp.	*Pantoea* spp.	*Proteus* spp.	*Pseudomonas* spp.	*Providencia* spp.	*Salmonella* spp.	*Serratia* spp.	*Shigella* spp.	*Stenotrophomonas maltophilia*	*Aspergillus* spp.	*Acremonium* spp.	*Alternaria* spp.	*Aureobasidium pullulans*	*Bravibactrium casei*	*Candida* spp.	*Cephalosporium* spp.	*Cercospora* spp.	*Cellulosimicrobium cellulans*	*Cladosporium* sp.	Exiguobacterium	*Fonsecaea* spp.	*Fusarium* spp.	*Gliocladium* spp.	*Geotricum* spp.	*Mucor* spp.	*Penicillium* spp.	*Rothiarhodotorula rubra Streptomyces* spp. *Trichosporon* spp. *Yeast* not *candida Yeast* spp. Unidentified
																																																														[41]
																																																														[42]
																																																														[49]
																																																														[56]
																																																														[30]
																																																														[36]
																																																														[19]
																																																														[37]
																																																														[23]
																																																														[13]
																																																														[57]
																																																														[34]
																																																														[22]
																																																														[48]
																																																														[25]
																																																														[47]
																																																														[39]
																																																														[12]
																																																														[14]
																																																														[46]
																																																														[50]
																																																														[15]
																																																														[24]
																																																														[27]
																																																														[28]
																																																														[29]
																																																														[35]
																																																														[43]
																																																														[21]
																																																														[18]
																																																														[17]

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
