# Peer review of "Highlighting the Microbial Contamination of the Dropper Tip and Cap of In-Use Eye Drops, the Associated Contributory Factors, and the Risk of Infection: A Past-30-Years Literature Review"

_pharmaceutics, 2022, doi:10.3390/pharmaceutics14102176_

Round 1

Reviewer 1 Report

Manuscript is good. But there are few corrections to be done:

1.     Enhance the resolution of Figure 1

2.      In Table 1 the studies should be arranged chronologically from current year to previous years.

1.      Conclusion is very short should be elaborated.

Author Response

Dear reviewer,

We would like to thank you for your time and effort and for your valuable comments.

  1.    Enhance the resolution of Figure 1: Done.
  2. In Table 1 the studies should be arranged chronologically from the current year to previous years: Done

       3.Conclusion is very short should be elaborated: The conclusion was edited, and additional inferences were added.

    " The dropper tip and the cap of in-use eye drops are sources of microbial contamination and a risk of infection and ocular injury in susceptible patients. Evidence showed that handling eye drops by patients, caregivers, and healthcare providers is a risk for dropper tip contamination. The other contributory factors of microbial contamination were inconsistent across the literature to reach a consensus of use and determine the true magnitude and the clinical impact of contamination of these eye drops sites. Many questions remain unanswered, such as the impact of the frequency of use, extended use beyond the recommended period, re-use of unit-dose formulations, including generated plastic waste, and the associated costs. New technologies offer a promising potential for securing the long-term sterility of in-use eye drops, and could also benefit from a demonstration of effectiveness in the framework of a well-defined study protocol."

Reviewer 2 Report

Dear Authors,

Firstly, I congratulate you for undertaking the herculean task of collecting the relevant literature by choosing to report on the microbial contamination of the dropper tip and cap, a topic that is highly relevant to eye care practitioners, in general. This review paper  merely reported the microbial contamination by categorizing it under different sections i.e., Preservative vs preservative free eye drops by use, by site of contamination, by type of various eye drops and isolated microorganisms.  Further, the authors identified the majors factors associated with microbial contamination by user type, by type of settings where the eye drops were used, duration of use and handling techniques. While this information seem to be relevant, perhaps the use of statistical analysis might have been insightful to determine the strength of reported associations. Discussion was reasonably interesting and I agree with authors recommendations.

Overall, in my opinion, the methods section can be reorganized avoiding repetition for better flow and comprehension.

My specific comments are as below:

Line 183 - 196:  I suggest tabulating this information based on the settings from which the microbial contamination  was reported.

Lines 205 - 217: Interesting observation about different contamination rates with different eye drops, does the authors know if it related to the frequency of application of drops as well?

Lines 235 - 248: For better comprehension, I suggest to restructure this write-up while reporting findings from similar study designs, also including the design of bottle as applicable if that information is available.

Line 281: Fertility? Do you mean sterility?

Lines 342-346: Are the reported differences in microbial contamination between different type of eye drops due to the differences in the frequency of application?

Lines 355 - 357: Can the authors draw inferences from this finding?

Line 658: ? typo

Lines 706 and 731: Are the authors referring to injury or infection?

General comments:

It is very confusing for the reader while comprehending two different sections that are based on  preservative and preservative free  3.1.1 and 3.3.2. Please reorganize.

Has there been any reports citing microbial contamination rates when the use of eye drops was stretched beyond recommended applications.

Tables: Please have the headers running through each page

Table 4 is missing.

I suggest including figures where relevant to categorize the information.

Wishing the authors all the very best.

Thank you 

Author Response

Dear reviewer,

We would like to thank you for your time and efforts and for your valuable comments. Find enclosed, the answers to the addressed comments and suggestions.

Best regards,

Reviewer 3 Report

This excellent review by Iskandar et al. will fill a much-needed gap in the literature concerning rates, routes, and types of microbial transmission and risk of ocular infection via the dropper tip and/or bottle cap.  This comprehensive and thorough review conveniently draws together findings reported sporadically within the literature over the past 30 years, and should serve as a rich and much-cited source by clinicians and vision scientists with an interest in such matters.  The manuscript is well-written in general.  I have only a few minor comments, many relating to the use of English by the authors.

Materials and Methods (2.1 Context)  Line 101:  This final sentence would read better (to a native English speaker) if the word 'setting' were placed immediately following the words 'home-based' (i.e. ..... long term facilities, primary care clinics, and the home-based setting.)

Figure 2 & Table 3.  Please increase the size of the font.  As it stands, the figure is hard to read for a middle-aged presbyope such as myself, even with suitable spectacle correction. 

Tables 1 & 2.  If feasible, it would be nice to have the header panel for each table reproduced on every page where each table appears.

Section 3.1.1. (Lines 173-221, and beyond) Preserved eye drops collected from inpatient and outpatient settings.  In many places in the paragraph under this heading, and in some subsequent paragraphs, many of the % figures given are seemingly unnecessarily surrounded by parentheses.  Perhaps in French this grammatical construct is de rigueur, but to a native English speaker, this will seem odd.   

Section 3.2. Contaminated eye drop sites.  (Line 281).  Again, to a native English speaker, the use of the word 'fertility' in the context meant by the authors is a little odd.  It would read better if the words 'propensity for microbial growth' were substituted here (i.e. Most ophthalmic solutions are characterized by low propensity for microbial growth, even if preservative-free, ...) 

Discussion.  Lines 746 & 747.  'tones' should be written as 'tonnes'.   

Author Response

Dear reviewer,

We would like to thank you for your time and effort and for your valuable comments.

-Materials and Methods (2.1 Context) Line 101:  This final sentence would read better (to a native English speaker) if the word 'setting' were placed immediately following the words 'home-based' (i.e. ..... long term facilities, primary care clinics, and the home-based setting.): Added

-Figure 2 & Table 3.  Please increase the size of the font.  As it stands, the figure is hard to read for a middle-aged presbyope such as myself, even with suitable spectacle correction: done with much appreciation and great respect 

-Tables 1 & 2.  If feasible, it would be nice to have the header panel for each table reproduced on every page where each table appears: done

-Section 3.1.1. (Lines 173-221, and beyond) Preserved eye drops collected from inpatient and outpatient settings.  In many places in the paragraph under this heading, and in some subsequent paragraphs, many of the % figures given are seemingly unnecessarily surrounded by parentheses.  Perhaps in French this grammatical construct is de rigueur, but to a native English speaker, this will seem odd: done   

-Section 3.2. Contaminated eye drop sites.  (Line 281).  Again, to a native English speaker, the use of the word 'fertility' in the context meant by the authors is a little odd.  It would read better if the words 'propensity for microbial growth' were substituted here (i.e. Most ophthalmic solutions are characterized by low propensity for microbial growth, even if preservative-free, ...) : Changed

-Discussion.  Lines 746 & 747.  'tones' should be written as 'tonnes': done